# Disentangling Exploitative and Interference Competition on Forest Dwelling Salamanders

**DOI:** 10.3390/ani13122003

**Published:** 2023-06-15

**Authors:** Giacomo Rosa, Sebastiano Salvidio, Andrea Costa

**Affiliations:** Department of Earth, Environment and Life Sciences (DISTAV), University of Genova, Corso Europa 26, 16126 Genova, Italy

**Keywords:** exploitative competition, interference competition, multi-species N-mixture models, salamanders, temporal niche partitioning

## Abstract

**Simple Summary:**

Exploitative competition and interference competition differ in the way access to resources is modulated by a competitor. Exploitative competition implies resource depletion and usually produces spatial segregation, while interference competition is independent from resource availability and can result in temporal niche partitioning. Here, we inferred the presence of these two patterns of competition on a two-salamander system in Northern Italy. We found evidence supporting interference competition and temporal niche partitioning.

**Abstract:**

Exploitative competition and interference competition differ in the way access to resources is modulated by a competitor. Exploitative competition implies resource depletion and usually produces spatial segregation, while interference competition is independent from resource availability and can result in temporal niche partitioning. Our aim is to infer the presence of spatial or temporal niche partitioning on a two-species system of terrestrial salamanders in Northern Italy: *Speleomantes strinatii* and *Salamandrina perspicillata*. We conducted 3 repeated surveys on 26 plots in spring 2018, on a sampling site where both species are present. We modelled count data with N-mixture models accounting for directional interactions on both abundance and detection process. In this way we were able to disentangle the effect of competitive interaction on the spatial scale, i.e., local abundance, and from the temporal scale, i.e., surface activity. We found strong evidence supporting the presence of temporal niche partitioning, consistent with interference competition. At the same time, no evidence of spatial segregation has been observed.

## 1. Introduction

Identifying which ecological processes influence the occurrence and local abundance of a species is of fundamental interest among ecologists and conservationists. Local abundance for a given species is expected to reach its maximum when environmental features correspond to the optimum of its ecological requirements [1,2]. However, the local rise in abundance or density of a species as resources or suitability of a given area increase is counterbalanced and limited by biotic interactions with other organisms that share the same ecological niche, usually through density-dependent competitive interactions [3,4].

Even though the study of competition laid its root early in the field of ecology [5], nowadays, the topic is still alive and debated, and the consequences of competitive interactions are still attracting interest among the scientific community. This happens at the theoretical [6,7,8] and practical level [9,10]. Competitive interactions can occur at the inter- or intra-specific level and can involve various niche axes and mechanisms. For instance, organisms compete directly for mates (e.g., [11,12]) and physical space, such as habitat or shelter use [13], and indirectly for food resources [14] or other niches, such as call space [15]. Regardless of the mechanism or origin of competitive interactions, they usually occur when one species prevents the access of its competitors to a shared resource. This exclusion from use of a resource usually takes place in two different ways: exploitative competition or interference competition [13,16,17]. These two types of competition differ substantially in the way they affect the access of one species to a given resource. In the case of exploitative competition, some organisms make better use of the same limiting resource, directly reducing the availability of the shared resource for other individuals [18]. By contrast, interference competition occurs when one organism actively prevents another organism from accessing a resource, independently from its availability, for instance by means of territorial or aggressive behaviors [13,16].

Both types of competition are assumed to have significant effects on population dynamics, local abundance, community assembly, and species coexistence [5,19,20]. Indeed, exploitative competition can either lead to the exclusion of a species from a given habitat [18], resulting in a density reduction of one competitor, or can trigger resource partitioning among different niche axes [21,22,23]. Interference competition can trigger competitive exclusion [3], causing behavioral changes, for instance by shifting species activity patterns [13,17]. One example of interference competition is the shift to crepuscular activity of a species that avoids the presence of a more nocturnal competitor [17].

Likewise, different kinds of competitive interactions can result in different layouts of niche and resource partitioning, minimizing niche overlap and competitive exclusion [13]. For instance, when two species or organisms segregate over different microhabitats, we observe spatial niche partitioning; conversely, when organisms exploit the same set of resources but in different time frames, we observe a partitioning along the temporal niche [2,24]. In this respect, exploitative and interference competition are also hypothesized to have different outcomes for what concern spatial or temporal segregation/partitioning. Exploitative competition has been found to have an effect on the local abundance or spatial displacement of one or both competitors (e.g., [21]). This is because of a density-dependent effect on limited resources, in which the under-performing competitor must rely on the leftover resource by the best-performing competitor [25,26]. Conversely, interference competition usually produces niche shift patterns that are consistent with temporal niche partitioning, i.e., when species access the same resource in different timeframes in order to reduce their interactions with the competitor (e.g., [17,27,28]).

In the present study, our aim was to test the occurrence of exploitative or interference competition in a two-species system of forest dwelling salamanders, the Strinati’s cave salamander (*Speleomantes strinatii*) and the northern spectacled salamander (*Salamandrina perspicillata*). These two salamanders are found syntopically in NW Italy, where they rely on the same set of trophic resources, particularly soil-dwelling invertebrates [29,30]. We used binomial N-mixture models that account for directional biotic interaction on both the local abundance and the timing of surface activity to alternatively test the occurrence of spatial niche partitioning or temporal niche partitioning, after modelling abundance and activity on environmental covariates. We expected that (i) in the case of exploitative competition, we should observe a negative interaction of one species on the local abundance of the other, this being a negative density-dependent effect; or (ii) in the case of interference competition we should observe a negative interaction of one species on the surface activity of the other, this outcome representing niche partitioning on the temporal axis.

## 2. Materials and Methods

### 2.1. Study Framework

To test for the presence of exploitative or interference competition, we randomly selected 26 permanent plots under a meta-population design [31,32], in one site in Northern Italy where both salamanders are present. We counted salamanders in all plots during three repeated surveys over a short time period. We then used count data to model abundance and activity interaction occurring between the species, using a two-species N-mixture model [31], while incorporating environmental covariates [33,34,35,36,37,38].

### 2.2. Study Species

The first focal species is the Strinati’s cave salamander *Speleomantes strinatii*, a plethodontid salamander with a maximum total length of about 128 mm. The second focal species is the northern spectacled salamander *Salamandrina perspicillata*, that belongs to family Salamandridae; this salamander is smaller, not exceeding 100 mm in total length. Strinati’s cave salamanders possess direct development and are fully terrestrial, while northern spectacled salamanders have a biphasic life cycle, lay eggs in the water, and become fully terrestrial after metamorphosis, with only reproductive females entering the water for egg deposition [39,40]. Adults of both species are lungless (*S. strinatii*) or have vestigial lungs (*S. perspicillata*), and gas exchange occurs mainly through their skin. Both species are usually found in the talus and the leaf litter of mixed broadleaved woodlands, and their surface activity is mainly regulated by the amount of rain or soil moisture [29,37,41]. In forest environments, the diet of *S. strinatii* and *S. perspicillata* has been extensively studied, by means of stomach flushing: both species feed on a large number of invertebrate taxa and share many trophic resources [29,30]. However, at the population level *S. strinatii* has been described as a generalist predator [29], while *S. perspicillata* is a specialized predator on springtails and mites [30].

### 2.3. Study Site

The study area is located in North-Western Italy, on the Apennine Mountain range of the Piemonte region, where the two studied species occurs sympatrically and are often found syntopically. The sampling area is located at an altitude of 600 m a.s.l. in the municipality of Mongiardino Ligure (44°38′24″ N; 9°03′00″ E), Alessandria province. The site is crossed by a first order Apennine stream and is characterized by a Supra–Mediterranean mixed deciduous forest, dominated by European Oak (*Quercus pubescens* Willd., 1805; [42]). We selected, individually marked, and GPS-positioned 26 square plots in the proximity of the stream banks, each one measuring 30 m^2^ (5.5 m side). The minimum distance among plots was about 20 m.

### 2.4. Salamanders’ Sampling

In spring 2018 (16 April–25 May), the same observer visited all sites three times during daytime (8–11 a.m.) and in favorable weather conditions for salamander’s activity (e.g., during or immediately after light rain). During each survey the observer searched for salamanders within each plot for four minutes [37,43], checking the leaf litter, inspecting rock crevices with a flashlight, and lifting temporary shelters such as superficial rocks and dead wood in search of salamanders. Since both species are known to be active during or immediately after rain and considering that inactive individuals retreat underground and are usually unavailable for sampling, we considered both individuals encountered on the forest floor and those found under temporary shelters as active on the surface [37,38,41]. We assigned each individual to one of the two study species and their abundance in each plot recorded.

### 2.5. Environmental Covariates

Within each marked plot at a depth of 20 cm, we obtained five measurements of soil relative humidity using a digital moisture meter (Extech MO750). All measurements were obtained on the same day (20 April 2018), 4 days following a 50 mm precipitation event. The average of these five measurements was considered as a plot-specific proxy, describing the soil moisture retention potential (MOIST). From a digital elevation model (DEM; 20 m mesh size) of the study area, we calculated two covariates: the duration of direct insolation (INSOL), expressed in hours, and the topographic position index (TPI). This latter index expresses the topographic position of each cell within the landscape, assuming positive values for cells located on ridges or hilltops and negative values for cells located in depressions [44]. Moreover, for each sampling session, we recorded the day of the year (DAY; i.e., the continuous count of the number of days beginning each year from 1 January). We obtained the temperature of the survey (TEMP) and the accumulated rain in the 72 h prior to sampling (RAIN) from local weather stations. We conducted terrain analyses with software SAGA 7 [45].

### 2.6. Data Analysis

We analyzed our repeated count data of *S. strinatii* and *S. perspicillata* using a co-abundance formulation of the static binomial N-mixture model of Royle [31], accounting for directional biotic interactions [32], both on the abundance of salamanders and on their activity pattern. Binomial N-mixture models estimate the latent abundance state (*N*) at site *i* (*Ni*), assuming *Ni~Poisson(λ)*, where *λ* is the expected abundance over all sites, by using repeated counts *C* at site *i* during survey *j* (*Cij*) to estimate individual detection probability *p*, assuming *Cij|Ni~Binomial(Ni,p)*. Both parameters can be modelled as a function of environmental covariates trough a *log* or *logit* link, respectively. In order to model the possible effect of exploitative competition on local abundance, we stacked two N-mixture models, using the latent abundance of the larger species (i.e., *S. strinatii*) as a covariate in the abundance model of *S. perspicillata* [34,35,36]. Therefore, we added a co-abundance interaction effect term (*γ*) to the model [33] to estimate the overall effect of the abundance of Strinati’s cave salamanders on the abundance of the northern spectacled salamander [32]. Likewise, to model the possible presence of interference competition resulting from reduced activity of the smaller species when the larger one is active, we considered the detection probability *p* of both species to be a rough proxy of species’ surface activity. This approach is widely shared and often successfully used in ecological studies assessing species interactions [37,38,46]. Therefore, we added another interaction term (*ε*) on the detection process, using the detection probability *pij* of *S. strinatii* as a covariate on the detection of *S. perpicillata*. Prior to building our model, we standardized covariates and checked them for collinearity, considering a cut-off for inclusion of Pearson r < 0.7 [47,48]. We modelled the detection process of two species as follows: [1; *S. strinatii*] *logit*(*pij_Ss_*) = *α*0 + *α*1**DAYij* + *α*2**TEMPij* + *α*3**RAINij* + *δij*
[2; *S. perspicillata*] *logit*(*pij_Sp_*) = *α*0 + *α*1**DAYij* + *α*2**TEMPij* + *α*3**RAINij* + *ε*pij_Ss_* + *δij*
where the subscripts *Ss* and *Sp* stand for *S. strinatii* and *S. perspicillata,* respectively, *α*0 is the intercept, *α*1*–α*3 are covariate effects, *ε* is the directional interaction term on the activity between the two species, and *δ* is a normally distributed random effect that accounts for possible over dispersion in the detection process [49]. Similarly, we modelled the abundances of the two species as:[3; *S. strinatii*] *log* (*NiSs*) = *β*0 *+ β*1**MOISTi + β*2**INSOLi + β*3**TPIi + ηi*
[4; *S. perspicillata*] *log* (*NiSp*) = *β*0 *+ β*1**MOISTi + β*2**INSOLi + β*3**TPIi + γ*NiSs + ηi*
where the subscripts *Ss* and *Sp* stand for *S. strinatii* and *S. perspicillata,* respectively, *β*0 is the intercept, *β*1–*β*3 are covariate effects, *Ni* is the latent abundance of adults at site *i*, *γ* is the co-abundance effect of adults on juveniles, and *η* is a normally distributed site-level random effect [49]. We estimated model parameters using a Bayesian approach with Markov chain Monte–Carlo methods, using uninformative priors. We ran three chains, each one with an adaptive phase of 10,000 iterations, followed by 350,000 iterations, discarding the first 50,000 as a burn-in and thinning by 100. We considered that chains reached convergence when the Gelman–Rubin statistic (R-hat) was <1.1 [50]. We used the 90% highest density interval of the posterior distribution as a credible interval (CRI), and considered covariates on both abundance, detection, and biotic interaction to have a significant effect when CRI did not cross the zero. In order to assess model fit [51,52,53], we employed posterior predictive checks based on *χ^2^* statistics as a measure of the discrepancy between observed and simulated data and calculated a Bayesian *p*-value accordingly [49]. Analyses were conducted calling program JAGS (V4.3.0; [54]) from the R environment with package “JagsUI” (V1.5.1; [55]).

## 3. Results

During our sampling, we counted 96 salamanders: 61 *S. strinatii* and 35 *S. perspicillata*. For both species, the co-abundance N-mixture model was in a good fit (posterior predictive checks: Bayesian *p*-value for *S. strinatii* = 0.46, and for *S. perspicillata* = 0.51). For all monitored parameters, convergence resulted successful (maximum R-hat = 1.059). The complete list of parameters’ estimates, along with their respective 90% CRI, for both species, is reported in Table 1.

The two species showed different detection probabilities, with *S. strinatii*, the larger species, being more active on the forest floor (*p* = 0.43; CRI = 0.21–0.66) when compared to *S. perspicillata* (*p* = 0.32; CRI = 0.08–0.62). None of the survey covariates had a clear effect on *p* (CRI crossed zero, in all cases), although there was an 89.7% probability that TEMP had a negative effect on the detection of *S. perspicillata* (Pd, posterior probability of direction; Table 1). The two species also showed different per-site abundances, with *S. strinatii* (*λ* = 0.70; CRI = 0.25–1.15) being less abundant than *S. perspicillata* (*λ* = 2.10; CRI = 0.48–4.02).

Overall, the two species showed differential responses to the same set of environmental predictors, both for the ecological process and their detection/activity. Indeed, the abundance of *S. strinatii* was positively affected by MOIST (Figure 1) and negatively affected by INSOL (Figure 2), while abundance of *S. perspicillata* was only negatively affected by INSOL (Figure 2).

Finally, for what concern the two interaction terms, the one on the surface activity of *S. perspicillata* was negative and significant (*ε* = −3.76; CRI = −7.66–−0.09), highlighting that the surface activity of *S. strinatii* has a strong negative effect on the activity of *S. perspicillata* (Figure 3). At the same time, the interaction term on local abundance was also negative (*γ* = −0.18), but its CRI largely crossed the zero (CRI = −0.56–0.20; pd = 79.9%; Figure 3).

## 4. Discussion

Identifying the ecological processes that impact the presence and population size of a species holds significant importance for ecologists and conservationists, since the maximum local abundance of a species is observed when environmental characteristics align with its ecological requirements. On the other hand, the increase in local abundance or density of a species due to improved resources or suitability of a particular area is constrained by biotic interactions with other organisms that partially overlap the same ecological niche. These interactions typically occur as density-dependent competitive interactions, which can balance and limit local abundance. These competitive interactions can arise within or between species, encompassing a range of niche axes and mechanisms. For instance, organisms engage in direct competition for mates and physical space, such as habitats or shelters, as well as indirect competition for food resources or other niches, such as call space. Irrespective of the mechanism or origin of these competitive interactions, they typically emerge when one species hinders its competitors’ access to a shared resource. This exclusion from resource utilization generally occurs through two distinct pathways: exploitative competition or interference competition. These two forms of competition differ significantly in their impact on a species’ access to a particular resource. Exploitative competition arises when certain organisms make more efficient use of the same limited resource, thus directly reducing its availability for other individuals. Conversely, interference competition takes place when one organism actively obstructs another organism’s access to a resource, regardless of its availability.

In this study, we found evidence that interference competition, rather than exploitative competition, is shaping the co-occurrence of *Speleomantes strinatii* (Strinati’s cave salamander) and *S. perspicillata* (northern spectacled salamander), after accounting for differential habitat requirements. That is, after modelling ecological requirements of the two species by means N-mixture models using appropriate covariates on both the abundance and detection processes, we still detected some significant interaction between the two species, which is attributable to a biological interaction [32,33,34,35,36]. Specifically, after accounting for environmental covariates, we found that the activity of *S. strinatii* negatively affected the surface activity of *S. perspicillata*, indicating that the observed difference in surface activity between the two species can be explained by temporal niche partitioning. This suggests that the two species avoided direct interactions on the forest floor by being active at different times. Conversely, we did not find any effect of Strinati’s cave salamander’s abundance on northern spectacled salamander’s abundance at the scale at which the study has been conducted. This indicates that resource depletion (i.e., exploitative competition) probably does not limit the local abundance of either species in the study area. The overall findings of this study align with previous research on interference competition, which has been observed to produce niche shift patterns consistent with temporal niche partitioning [17,27,28].

In contrast, our results did not provide support for the hypothesis of exploitative competition, which influences local abundance or spatial displacement of one or both competitors due to density-dependent effects on limited resources [21,25,26]. The lack of evidence for exploitative competition in this two-species system may imply that resource availability for these salamander species is not a major ecological limiting factor, or that they have developed mechanisms partitioning resources in a way that avoids direct competition [13]. Indeed, *S. perspicillata* and *S. strinatii* share some of the same trophic resources, but (i) display different trophic strategies [29,30], (ii) forage successfully in slightly different weather conditions [56], and (iii) when they both are active on the forest floor, *S. perspicillata* reduces the foraging intensity on the core prey items composing its diet, suggesting some kind of interference with *S. strinatii* [57].

Such niche differentiation likely relaxes competition for food, explaining why we did not detect exploitative effects. Temporal niche partitioning due to interference competition has been observed in several taxa, including amphibians [58], birds [59], and mammals [60]. For other forest salamanders, Jaeger and colleagues [58,61,62] found evidence of temporal niche partitioning at the intraspecific level in the eastern red-backed salamander (*Plethodon cinereus*), which was driven by interference competition for food resources and territories during the breeding season. However, interference competition has also been found to produce negative effects on the fitness and survival of competitors. Hairston et al. [63] found that interference competition between *P. cinereus* and *P. richmondi* resulted in reduced growth rates and survival of *P. richmondi* due to aggressive interactions with *P. cinereus*. Similarly, Petranka and Smith [64] found that interference competition between *P. cinereus* and *P. glutinosus* resulted in decreased survival and growth rates of both species.

*Salamandrina perspicillata* in the study site, despite being the outperformed species (i.e., its overall activity is reduced by the surface activity of *S. strinatii*), is roughly three times more abundant than *S. strinatii*. This is further evidence for the lack of a density-related exploitative competition on this two-species system, otherwise *S. perspicillata* abundance should be reduced. This also suggests the presence of behavioral strategies to minimize the overlap in their realized temporal niches. This may have been facilitated by their ability to successfully exploit similar prey resources at different times, thus minimizing the intensity of their competitive trophic interactions [2,24].

## 5. Conclusions 

Our results support the hypothesis that interference competition promotes temporal niche partitioning in terrestrial salamanders. By shifting their surface activity, *S. strinatii* and *S. perspicillata* can co-occur at high densities while minimizing competitive interactions. On the other hand, trophic niche differentiation, divergences in metabolism, prey selection or behavioral traits, together with a high prey abundance may help to prevent the emergence of competition for food resources and may promote species coexistence [65]. Nevertheless, further studies in different ecological situations and adopting different sampling frameworks specifically designed to model surface activity should be conducted to help clarify the role of temporal or spatial niche partitioning in shaping the coexistence of these salamander species.

## Figures and Tables

**Figure 1 animals-13-02003-f001:**
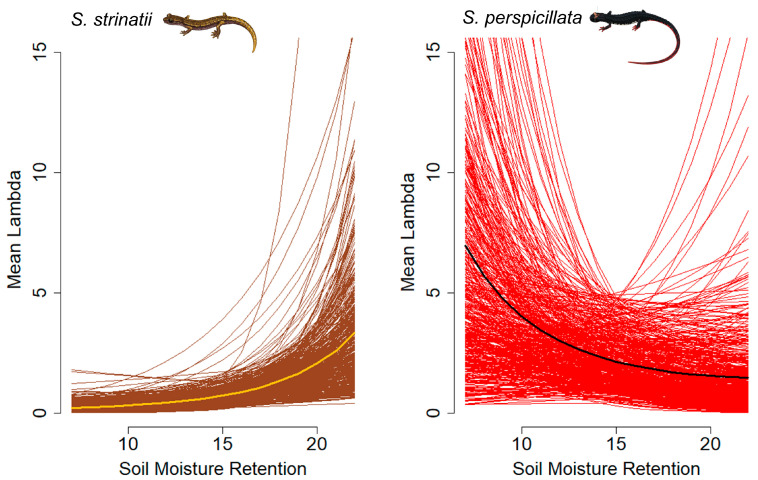
Plots showing the effect of the soil moisture retention potential (MOIST) on salamander abundance: *Speleomantes strinatii* on the **left** and *Salamandrina perspicillata* on the **right**. Bold lines represent the posterior mean, while the thin lines represent 500 random draws from the posterior distribution.

**Figure 2 animals-13-02003-f002:**
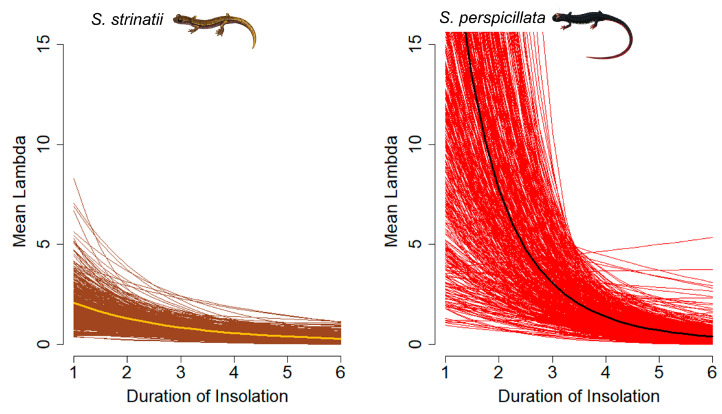
Plots showing the effect of duration of direct insolation (INSOL) on salamander abundance: *Speleomantes strinatii* on the **left** and *Salamandrina perspicillata* on the **right**. Bold lines represent the posterior mean, while the thin lines represent 500 random draws from the posterior distribution.

**Figure 3 animals-13-02003-f003:**
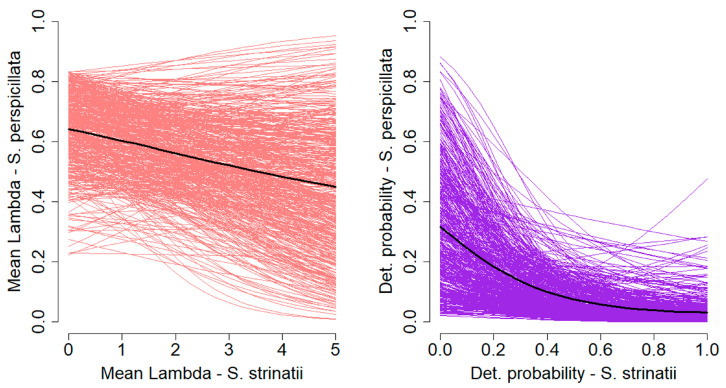
Plots showing the effect of surface activity (**left panel**; interaction terms *ε*) and abundance of *S. strinatii* (**right panel**; interaction term *γ*) on the surface activity and abundance of *S. perspicillata*. Bold lines represent the posterior mean, while the thin lines represent 500 random draws from the posterior distribution.

**Table 1 animals-13-02003-t001:** Complete list of the parameters estimated for the detection process, abundance, and interactions for both *S. strinatii* and *S. perspicillata*. Mean = mean obtained from the posterior distribution; 90% CRI = 90% highest density region of the posterior distribution; pd = probability of direction.

*Speleomantes strinatii*				*Salamandrina perspicillata*			
Parameter	Mean	90% CRI	pd	Parameter	Mean	90% CRI	pd
**Detection**				**Detection**			
Mean *p*	0.43	0.21–0.66	-	Mean *p*	0.32	0.08–0.62	–
DAY	0.04	−3.9–−4.3	50.7	DAY	−0.43	−4.85–3.61	57.0
TEMP	1.32	−1.7–4.4	76.6	TEMP	−2.59	−5.88–0.97	89.7
RAIN	1.33	−1.56–4.32	77.0	RAIN	−1.81	−5.2–1.37	82.0
**Abundance**				**Abundance**			
Mean *λ*	0.70	0.25–1.15	-	Mean *λ*	2.10	0.48–4.02	-
MOIST	0.88	0.25–1.57	98.7	MOIST	−0.48	−1.30–0.29	85.0
INSOL	−0.76	−1.31–−0.20	99.2	INSOL	−1.41	−2.34–−0.37	99.5
TPI	−1.11	−0.48–0.24	70.0	TPI	−0.33	−0.99–0.36	79.6
				**Interactions**			
				ε	−3.76	−7.66–−0.09	94.8
				γ	−0.18	−0.56–0.20	79.9

## Data Availability

The data presented in this study are available from the corresponding author upon reasonable request.

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
