# Peer review of "Disentangling Exploitative and Interference Competition on Forest Dwelling Salamanders"

_animals, 2023, doi:10.3390/ani13122003_

Round 1

Reviewer 1 Report

L 76-84: I think it is important here to provide more detail about overlap in diets (you keep it very general with soil-dwelling invertebrates). I also think it is important to indicate that both species are on the IUCN redlist as endangered, which would be a good rationale to find out how the populations are impacting each other.

Section 2.4 (L 120-127)

Why do you consider salamanders you found under cover objects (rocks and wood) or in crevices as being active? In other plethodontids, like P. cinereus and some desmognathines, active salamanders are only those in the leaf litter (or if surveying at night at the surface of the leaf litter). Lumping salamanders in the leaf litter and under cover objects similarly may not be appropriate and should be addressed with an explanation. Would it have been more appropriate to survey at night, assuming these salamanders are nocturnal?

L210-211: delete part of sentence from "although...to direction" since this wasn't significant.

L220: Do you mean MOIST instead of HUMID?

L220-221 (and table): Is the CRI correct for INSOL here? 

L 230-231: Delete from "negative...zero" because it is not significant?

L240-241: What was the temporal partitioning? Could you explain this a bit more.

Suggested grammatical changes

L1: Add "the" between "influence" and "occurrence"

L34: delete both commas, add "of a species" after "density" 

L35: Replace "Despite" with "Even though"

L35: Replace "at the dawn of" with "early in the field"

L38: Replace "drawbacks" with "consequences" or "impacts"

L40: Delete "either"

L41: replace first "and" with "or", delete "can"

L42: change "directly compete" to "compete directly", , relace comma in front of "physical space" with "or"

L45: delete "the"

L46: change "from a resource use" to "from use of a resource"

L47: change "and" to "or"

L49: add "some" in front of "organisms"

L50: change "affecting" to "reducing"

L52: change "from" to "of"

L55: add comma after "assembly"

L65: delete "a"

L66: delete comma after "resources" and add comma after "frames"

L69: add "an" between "have" and "effect"

L70: change "density dependent" to density-dependent"

L77: "Salamanders" should not be capitalized

L78: "Northern" should not be capitalized

L79: Change "sintopically" to "syntopically"

L80: Change "in particular to" to "particularly,"

L87: change "density dependent" to density-dependent", change "effects" to "effect"

L96: delete "interactions"

L99: Change "Strinati'" to "Strinati's"

L101: "Spectacled" should not be capitalized

L105: Replace "and" with "with", Change "enter" to "entering"

L106: Delete "devote", change "gas exchange mainly to" to "gas exchange occurs mainly through"

L111: Change "simpatrically" to "sympatrically"

L112: Add "the" between "of" and "Piemonte" 

L113: Change "sintopically" to "syntopically"

L116: Add "by" between "dominated" and "European"

L140: Change "cumulated" to "accumulated"

L203 and 204: Italicized genus species names

L253-254: Change "to partition their resources in such a way to aviod direct competition" to ", which partition their resources in a way that avoids direct competition"

 L257: replace "core preys" with "core prey items"

L262: Replace "what concerns" for "other"

L263: change "Red backed" to "eastern red-backed"

L264: change "to be" with "that was"

L272: Change "in" to "at"

L273: place parentheses around "i.e. its overall...S. strinatti)", add comma after "i.e."

L274: Delete "a" in front of "evidence"

L275: replace "on" with "in"

Author Response

L 76-84: I think it is important here to provide more detail about overlap in diets (you keep it very general with soil-dwelling invertebrates). I also think it is important to indicate that both species are on the IUCN redlist as endangered, which would be a good rationale to find out how the populations are impacting each other.

Authors: We agree with the Reviewer. Therefore we added the followinf paragraph in Materials and Methods section, to better detail the trophic ecology of the two species: “    In forest environments, the diet of S. strinatii and S. perspicillata has been extensively stud-ied, by means of stomach flushing: both species feed on a large number of invertebrate taxa and share many trophic resources [29,30]. However, at the population level S. strinatii has been described as a generalist predator [29], while S. perspicillata is a specialized predator on springtails and mites [30].”

Section 2.4 (L 120-127): Why do you consider salamanders you found under cover objects (rocks and wood) or in crevices as being active? In other plethodontids, like P. cinereus and some desmognathines, active salamanders are only those in the leaf litter (or if surveying at night at the surface of the leaf litter). Lumping salamanders in the leaf litter and under cover objects similarly may not be appropriate and should be addressed with an explanation. Would it have been more appropriate to survey at night, assuming these salamanders are nocturnal?

Authors: We totally acknowledge Reviewer’s comment. Now we give more details in this section of the manuscript and we clearly explain why we considered individuals under temporary shelters as “active” on the forest floor. We also acknowledge that nocturnal sampling could be appropriate, however we conducted sampling after rainfall or in moist conditions, when both species are known to be active also during the day. Indeed, in several studies conducted on both species, we found that surface activity and foraging activity is mainly driven by the accumulated rain prior sampling and by soil moisture conditions.   

L210-211: delete part of sentence from "although...to direction" since this wasn't significant.

Authors: Deleted as suggested

L220: Do you mean MOIST instead of HUMID?

Authors: Yes, it was a typo from a previous version of the manuscript, now corrected

L220-221 (and table): Is the CRI correct for INSOL here? 

Authors: Insol has a negative “significant” effect (CRI entirely on the left of the 0) for both species. As the Reviewer suggested a ‘-‘ was missing in the table, now corrected.

L 230-231: Delete from "negative...zero" because it is not significant?

Authors: we deleted the sentence as suggested.

L240-241: What was the temporal partitioning? Could you explain this a bit more.

Authors: we acknowledge Reviewer’s comment and expanded this concept as follows: “Specifically, after accounting for environmental covariates, we found that the activity of S. strinatii negatively affected the surface activity of S. perspicillata, indicating that the ob-served difference in surface activity between the two species can be explained by temporal niche partitioning. This suggests that the two species avoided direct interactions on the forest floor by being active at different times.”

Comments on the Quality of English Language Suggested grammatical changes

L1: Add "the" between "influence" and "occurrence"

Authors: Changed as suggested

L34: delete both commas, add "of a species" after "density"

 Authors: deleted

L35: Replace "Despite" with "Even though"

Authors: replaced

L35: Replace "at the dawn of" with "early in the field"

Authors: Replaced

L38: Replace "drawbacks" with "consequences" or "impacts"

Authors: Replaced

L40: Delete "either"

Authors: Deleted

L41: replace first "and" with "or", delete "can"

Authors: Changed as suggested

L42: change "directly compete" to "compete directly", , relace comma in front of "physical space" with "or"

Authors: Changed as suggested

L45: delete "the"

Authors: Deleted

L46: change "from a resource use" to "from use of a resource"

Authors: Changed

L47: change "and" to "or"

Authors: Changed

L49: add "some" in front of "organisms"

Authors: Added

L50: change "affecting" to "reducing"

Authors: Changed

L52: change "from" to "of"

Authors: Changed

L55: add comma after "assembly"

Authors: added a comma as suggested

L65: delete "a"

Authors: Deleted “a”

L66: delete comma after "resources" and add comma after "frames"

Authors: Deleted

L69: add "an" between "have" and "effect"

Authors: Added

L70: change "density dependent" to density-dependent"

Authors: Changed

L77: "Salamanders" should not be capitalized

Authors: Changed

L78: "Northern" should not be capitalized

Authors: Changed

L79: Change "sintopically" to "syntopically"

Authors: Changed

L80: Change "in particular to" to "particularly,"

Authors: Changed

L87: change "density dependent" to density-dependent", change "effects" to "effect"

Authors: Changed

L96: delete "interactions"

Authors: We specifically modellled interaction on abundance and activity, therefore we retained the word in this sentence.

L99: Change "Strinati'" to "Strinati's"

Authors: Changed

L101: "Spectacled" should not be capitalized

Authors: Changed

L105: Replace "and" with "with", Change "enter" to "entering"

Authors: Replaced

L106: Delete "devote", change "gas exchange mainly to" to "gas exchange occurs mainly through"

Authors: Changed

L111: Change "simpatrically" to "sympatrically"

Authors: Changed

L112: Add "the" between "of" and "Piemonte" 

Authors: Added

L113: Change "sintopically" to "syntopically"

Authors: Changed

L116: Add "by" between "dominated" and "European"

Authors: Added

L140: Change "cumulated" to "accumulated"

Authors: Changed

L203 and 204: Italicized genus species names

Authors: Changed

L253-254: Change "to partition their resources in such a way to aviod direct competition" to ", which partition their resources in a way that avoids direct competition"

Authors: Changed

 L257: replace "core preys" with "core prey items"

Authors: Changed

L262: Replace "what concerns" for "other"

Authors: Changed

L263: change "Red backed" to "eastern red-backed"

Authors: Changed

L264: change "to be" with "that was"

Authors: Changed

L272: Change "in" to "at"

Authors: Changed

L273: place parentheses around "i.e. its overall...S. strinatti)", add comma after "i.e."

Authors: Changed

L274: Delete "a" in front of "evidence"

Authors: Deleted

L275: replace "on" with "in"

Authors: Replaced

Reviewer 2 Report

The manuscript is clear and well presented. There are minor typographical issues and close proof-reading is still needed. For example, in the abstract, 

line 12: northern

line15: ..'is modulated by a competitor..'

I am not clear on what the required style for abbreviation of species where the genus names start with the same letter: in some journals, and I think most commonly, they would be abbreviated as Sp. strinatii and Sa. perspicillata, rather than both with S. Please check - I could not find this in the author instructions for Animals or the style guide for MDPI.

In line 220 the parameter 'HUMID' appears, not mentioned otherwise. Is this the same as MOIST and should it be here?

Given that the niche partitioning might be explained by environmental conditions alone, would you like to make a comment on what the limitations of the binomial N-mixture model is and what the likelihood is that it correctly represents real interactions?

Very good. A few typos, proof read carefully

Author Response

The manuscript is clear and well presented. There are minor typographical issues and close proof-reading is still needed. For example, in the abstract, 

line 12: northern

Authors: Changed as suggested

line15: ..'is modulated by a competitor..'

Authors: Changed

I am not clear on what the required style for abbreviation of species where the genus names start with the same letter: in some journals, and I think most commonly, they would be abbreviated as Sp. strinatii and Sa. perspicillata, rather than both with S. Please check - I could not find this in the author instructions for Animals or the style guide for MDPI.

Authors: we haven’t found any information on this specific issue.

In line 220 the parameter 'HUMID' appears, not mentioned otherwise. Is this the same as MOIST and should it be here?

Authors: It was a typo from a previous version of the manuscript, now corrected

Given that the niche partitioning might be explained by environmental conditions alone, would you like to make a comment on what the limitations of the binomial N-mixture model is and what the likelihood is that it correctly represents real interactions?

Authors: We acknowledge Reviewer’s comment and expanded a sentence in order to discuss this specific issue as follows: “In this study, we found evidence that interference competition, rather than exploita-tive competition, is shaping the co-occurrence of Speleomantes strinatii (Strinati’s cave sal-amander) and S. perspicillata (Northern spectacled salamander), after accounting for dif-ferential habitat requirements. That is, after modelling ecological requirements of the two species by means n-mixture models, using appropriate covariates on both the abundance and detection processes, we still detected some significant interaction between the two species, which is attributable to a biological interaction [32-36].”

Comments on the Quality of English Language :Very good. A few typos, proof read carefully

Reviewer 3 Report

The authors tested for the type of competition occuring between two species of salamanders that are found syntopically. They found evidence for interference competition as indicated by temporal niche partitioning.  I think the results are good basic research and are worthy of publication.

See attached pdf for minor in line comments.

I made a few edits and suggestions for edits. 

Author Response

 The authors tested for the type of competition occuring between two species of salamanders that are found syntopically. They found evidence for interference competition as indicated by temporal niche partitioning.  I think the results are good basic research and are worthy of publication.

See attached pdf for minor in the comments.

L122: Now we added time range of the samplings

L124: We reference our methods in order to inform readers that this methodology has already been applied and evaluated/validated in previous studies.

L140: We moved 1st before January

L200: We acknowledge Reviewer’s comment and changed the sentence for a better clarity.

L220: We replaces “resulted to be” with “was” as suggested by the Reviewer

L233: We acknowledge Reviewer’s comment and slightly changed Figure 1, Figure 2 and Figure 3 captions to make them clearer.

L242: We acknowledge reviewer’s comment and gave scientific name together with common name at the beginning of the section to remind the reader which is which salamander.   

L249: Removed “has been previously found”

L274: Removed “a”

Comments on the Quality of English Language: I made a few edits and suggestions for edits. 

Round 2

Reviewer 1 Report

L 12: change "this" to "these"

L12: change "two salamanders system" to "two-salamander system"

Line 31: remove comma

L40: Chang "This happens both at the theoretical and at the practical level" to "This happens at the theoretical and practical level"

Line 43: remove comma before "physical space" and replace with the word "and"

Line 52: use "from" not "of" (in front of "accessing"

Line 64: to avoid teleology, change "to minimuze niche overlap" to "minimizing niche overlap"

L81: delete "and"

L82: delete "to"

L84:delete comma after "activity"

L86: delete comma after "that"

L88: change "effects" to "effect"

L94: delete comma

L102: change to "northern spectacled salamander"

L105: I don't believe northern should be capitalized in this journal

L107: add comma after lungless

L127: change "from the 16th of April to the 25th of May"  to "16 April - 25 May"

L128: remove comma after "8-11 hrs" and replace with "and"

L131-136: this is a better description, thank you.

L131: add comma after "flashlight"

L 132: delete comma

L136: change "individuals" to "individual"

L141: Change "20th April 2018" to "20 April 2018"

L147: change "hill-tops" to "hilltops"

L147: delete comma

L149: add comma after "i.e."

L150: change "1st January" to "1 January"

L166: I don't believe northern needs to be capitalized

Line 167: Change "on a" to "from"

Line 195: change "un-informative" to "uninformative"

L200: add comma after "detection"

L210: change "resulted" to "was"

L217: would it be helpful to include after "S. strinatii" ", the larger species,"

L225-226: change "salamanders" to "salamanders'"

L230: Change "S. strinatii abundance" to "abundance of S. strinatii" same for S. perspicillata on L 232

L235: change "salamanders" to "salamanders'"

L245: change "of S. strinatii surface activity (...) and abundance (...)" to "of surface activity (...) and abundance (...) of S. strinatii"

L251: is "Northern" supposed to be capitalized?

L253: change "n-mixture" to "N-mixture"

L253: delete comma after "models"

L261: change "Strinati's cave salamander" to Strinati's cave salamander's"

L261: use "northern spectacled salamander's" instead of "spectacled salamander's"

L268: remove paragraph break

L273: replace "mechanisms, which partition their resources in a way that avoids direct" to "mechanisms partitioning resources that avoids direct"

L274" should this say "share some of the same trophic resources"

L 283: change "intra-specific" to "intraspecific"

L284: change "red backed" to "red-backed"

English is fine with a few minor corrections.

Author Response

We followed all Reviewer's suggestions and amended the manuscript accordingly:

L 12: change "this" to "these"

Authors: changed as suggested

L12: change "two salamanders system" to "two-salamander system"

Authors: changed as suggested

Line 31: remove comma

Authors: changed as suggested

L40: Chang "This happens both at the theoretical and at the practical level" to "This happens at the theoretical and practical level"

Authors: changed as suggested

Line 43: remove comma before "physical space" and replace with the word "and"

Authors: changed as suggested

Line 52: use "from" not "of" (in front of "accessing"

Authors: changed as suggested

Line 64: to avoid teleology, change "to minimuze niche overlap" to "minimizing niche overlap"

Authors: changed as suggested

L81: delete "and"

Authors: changed as suggested

L82: delete "to"

Authors: changed as suggested

L84:delete comma after "activity"

Authors: changed as suggested

L86: delete comma after "that"

Authors: changed as suggested

L88: change "effects" to "effect"

Authors: changed as suggested

L94: delete comma

Authors: changed as suggested

L102: change to "northern spectacled salamander"

Authors: changed as suggested

L105: I don't believe northern should be capitalized in this journa

Authors: changed as suggested

L107: add comma after lungless

Authors: changed as suggested

L127: change "from the 16th of April to the 25th of May"  to "16 April - 25 May"

Authors: changed as suggested

L128: remove comma after "8-11 hrs" and replace with "and"

Authors: changed as suggested

L131-136: this is a better description, thank you.

Authors: changed as suggested

L131: add comma after "flashlight"

Authors: changed as suggested

L 132: delete comma

Authors: changed as suggested

L136: change "individuals" to "individual"

Authors: changed as suggested

L141: Change "20th April 2018" to "20 April 2018"

Authors: changed as suggested

L147: change "hill-tops" to "hilltops"

Authors: changed as suggested

L147: delete comma

Authors: changed as suggested

L149: add comma after "i.e."

Authors: changed as suggested

L150: change "1st January" to "1 January"

Authors: changed as suggested

L166: I don't believe northern needs to be capitalized

Authors: changed as suggested

Line 167: Change "on a" to "from"

Authors: changed as suggested

Line 195: change "un-informative" to "uninformative"

Authors: changed as suggested

L200: add comma after "detection"

Authors: changed as suggested

L210: change "resulted" to "was"

Authors: changed as suggested

L217: would it be helpful to include after "S. strinatii" ", the larger species,"

Authors: changed as suggested

L225-226: change "salamanders" to "salamanders'"

Authors: changed as suggested

L230: Change "S. strinatii abundance" to "abundance of S. strinatii" same for S. perspicillata on L 232

Authors: changed as suggested

L235: change "salamanders" to "salamanders'"

Authors: changed as suggested

L245: change "of S. strinatii surface activity (...) and abundance (...)" to "of surface activity (...) and abundance (...) of S. strinatii"

Authors: changed as suggested

L251: is "Northern" supposed to be capitalized?

Authors: changed as suggested

L253: change "n-mixture" to "N-mixture"

Authors: changed as suggested

L253: delete comma after "models"

Authors: changed as suggested

L261: change "Strinati's cave salamander" to Strinati's cave salamander's"

Authors: changed as suggested

L261: use "northern spectacled salamander's" instead of "spectacled salamander's"

Authors: changed as suggested

L268: remove paragraph break

Authors: changed as suggested

L273: replace "mechanisms, which partition their resources in a way that avoids direct" to "mechanisms partitioning resources that avoids direct"

Authors: changed as suggested

L274" should this say "share some of the same trophic resources

 Authors: changed as suggested

L 283: change "intra-specific" to "intraspecific"

Authors: changed as suggested

L284: change "red backed" to "red-backed"

Authors: changed as suggested